# Chronic social isolation reduces 5-HT neuronal activity via upregulated SK3 calcium-activated potassium channels

Derya Sargin[1], David K Oliver[1], Evelyn K Lambe[1,2,3]*

[1]Department of Physiology, University of Toronto, Toronto, Canada; [2]Department of Obstetrics and Gynaecology, University of Toronto, Toronto, Canada; [3]Department of Psychiatry, University of Toronto, Toronto, Canada

**Abstract** The activity of serotonin (5-HT) neurons is critical for mood regulation. In a mouse model of chronic social isolation, a known risk factor for depressive illness, we show that 5-HT neurons in the dorsal raphe nucleus are less responsive to stimulation. Probing the responsible cellular mechanisms pinpoints a disturbance in the expression and function of small-conductance $Ca^{2+}$-activated $K^+$ (SK) channels and reveals an important role for both SK2 and SK3 channels in normal regulation of 5-HT neuronal excitability. Chronic social isolation renders 5-HT neurons insensitive to SK2 blockade, however inhibition of the upregulated SK3 channels restores normal excitability. In vivo, we demonstrate that inhibiting SK channels normalizes chronic social isolation-induced anxiety/depressive-like behaviors. Our experiments reveal a causal link for the first time between SK channel dysregulation and 5-HT neuron activity in a lifelong stress paradigm, suggesting these channels as targets for the development of novel therapies for mood disorders.

*For correspondence: evelyn.lambe@utoronto.ca

Competing interests: The authors declare that no competing interests exist.

## Introduction

Major depression is a prevalent and debilitating disease for which standard treatments remain ineffective. Social isolation has long been implicated as a risk factor for depression in humans (*Cacioppo et al., 2002,2006*; *Holt-Lunstad et al., 2010*) and induces anxiety- and depressive-like behaviors in rodents (*Koike et al., 2009*; *Wallace et al., 2009*; *Dang et al., 2015*; *Shimizu et al., 2016*; for review see *Fone and Porkess, 2008*; *Lukkes et al., 2009*). The most frequently prescribed medications for depression and anxiety disorders target the serotonin (5-HT) producing neurons (*Blier et al., 1990*; *Nutt, 2005*), the majority of which are located in the dorsal raphe nucleus (DRN) (*Descarries et al., 1982*). The activity of 5-HT neurons is highly vulnerable to stress (*Lira et al., 2003*; *Bambico et al., 2009*; *Espallergues et al., 2012*; *Challis et al., 2013*) and is critical for depressive-like, anxiogenic, and reward-associated behaviors (*Liu et al., 2014*; *Teissier et al., 2015*; *Urban et al., 2016*). Interestingly, social isolation in rodents has been shown to affect endogenous 5-HT release and 5-HT turnover in postsynaptic areas (*Heidbreder et al., 2000*; *Muchimapura et al., 2002*, *2003*). However, it is unknown how social isolation affects the activity of DRN 5-HT neurons themselves. Identification of changes in the activity of these neurons may uncover novel therapeutic targets for depression and anxiety disorders.

Here, we show that chronic social isolation leads to a reduction in the excitability of DRN 5-HT neurons. Their firing activity to optical, electrophysiological, and neuromodulatory stimulation are all reduced after social isolation. Specifically, we have identified that the reduction in the firing activity of 5-HT neurons results from alterations in the function and expression of small-conductance $Ca^{2+}$-activated $K^+$ (SK) channels. Furthermore, inhibition of SK channels normalizes the activity of 5-HT neurons and restores the behavioral deficits observed after chronic social isolation.

**eLife digest** Major depressive disorder is a common and debilitating disease that interferes with the afflicted person's everyday life. While some patients do benefit from antidepressant treatments, these medications need to be taken for several weeks before they become effective. Still, a large proportion of patients do not recover fully and some do not respond at all to the existing treatments. As a result, there is a need to find new and more effective treatments for depression.

The most widely used antidepressant drugs target the chemical messenger or neurotransmitter called serotonin. The majority of nerve cells that produce serotonin are located in a region deep in the brain known as the dorsal raphe nucleus. When active, these nerve cells release serotonin; this in turn controls the cells' own activity as well as the activity of a large number of connected nerve cells located throughout the brain. Any disruption in this system will have a widespread impact and can potentially increase the risk of disturbed moods. However, it was not exactly clear what alters the activity of serotonin-producing nerve cells in depression.

Now, Sargin et al. have identified a previously unknown mechanism that underlies changes to the activity of serotonin-producing nerve cells. Keeping mice isolated for a prolonged period elicits the symptoms of depression. Sargin et al. found that serotonin-producing nerve cells were dramatically less active in isolated mice and that a specific type of ion channel protein (the SK3 channel) was more abundant in these nerve cells. A higher amount of this channel inhibits the activity of nerve cells. Blocking these inhibitory SK3 channels (using a drug that can be obtained from bee venom) restored normal activity in the serotonin-producing cells. Moreover, this treatment alleviated the depressive symptoms of the isolated mice.

The findings of Sargin et al. suggest a new way to treat the symptoms of depression. Yet to translate them into an accessible treatment for patients, future work will be required to develop drugs that can specifically and potently target the affected channel.

## Results

### Social isolation reduces excitability of 5-HT neurons

To examine how chronic social isolation affects DRN 5-HT neurons, we performed whole-cell electrophysiology on acute slices from mice expressing ChR2-EYFP under *Tph2* promoter (*Zhao et al., 2011*) (*Figure 1A*). Mice were single-housed after weaning for $\geq$7 weeks and group-housed littermates were used as controls. A striking difference was observed in 5-HT neuronal excitability to depolarizing current steps, with significantly fewer action potentials generated in 5-HT neurons from single-housed mice (*Figure 1B–C*). This difference did not arise from altered membrane characteristics (*Figure 1—figure supplement 1A–C*) nor altered GABA tone, as probed with picrotoxin and CGP52432 used to block $GABA_A$ and $GABA_B$ receptors, respectively (*Figure 1—figure supplement 1D*). The inhibitory current responses to the $5\text{-}HT_{1A}$ receptor agonist 5-CT were comparable between the groups (*Figure 1—figure supplement 2A–B*). Furthermore, the social isolation-induced difference in the intrinsic excitability of 5-HT neurons persisted in the presence of $5\text{-}HT_{1A}$ receptor blockade with WAY100635, indicating that this change does not result from altered serotonergic tone nor altered $5\text{-}HT_{1A}$ receptors (*Figure 1—figure supplement 2C–E*).

To investigate further the social isolation-induced changes in 5-HT neuron excitability, we evoked action potentials optogenetically (20 Hz blue light). Firing frequency of 5-HT neurons from single-housed mice in response to light was significantly reduced compared to those from group-housed littermates (*Figure 1D–E*). In contrast to optical stimulation, we were able to activate 5-HT neurons further using 20 Hz strong electrophysiological stimulation with current input >500 pA (with doublets in some neurons from group-housed mice). Again, 5-HT neurons from group-housed mice responded at a significantly higher frequency than those of single-housed mice (*Figure 1F–G*). Overall, chronic social isolation has led to a reduction in the excitability of 5-HT neurons rendering them less responsive to both optical and electrophysiological stimulations.

DRN 5-HT neurons receive numerous inputs (*Dorocic et al., 2014*; *Ogawa et al., 2014*; *Weissbourd et al., 2014*) that release excitatory neuromodulators such as orexins that are important

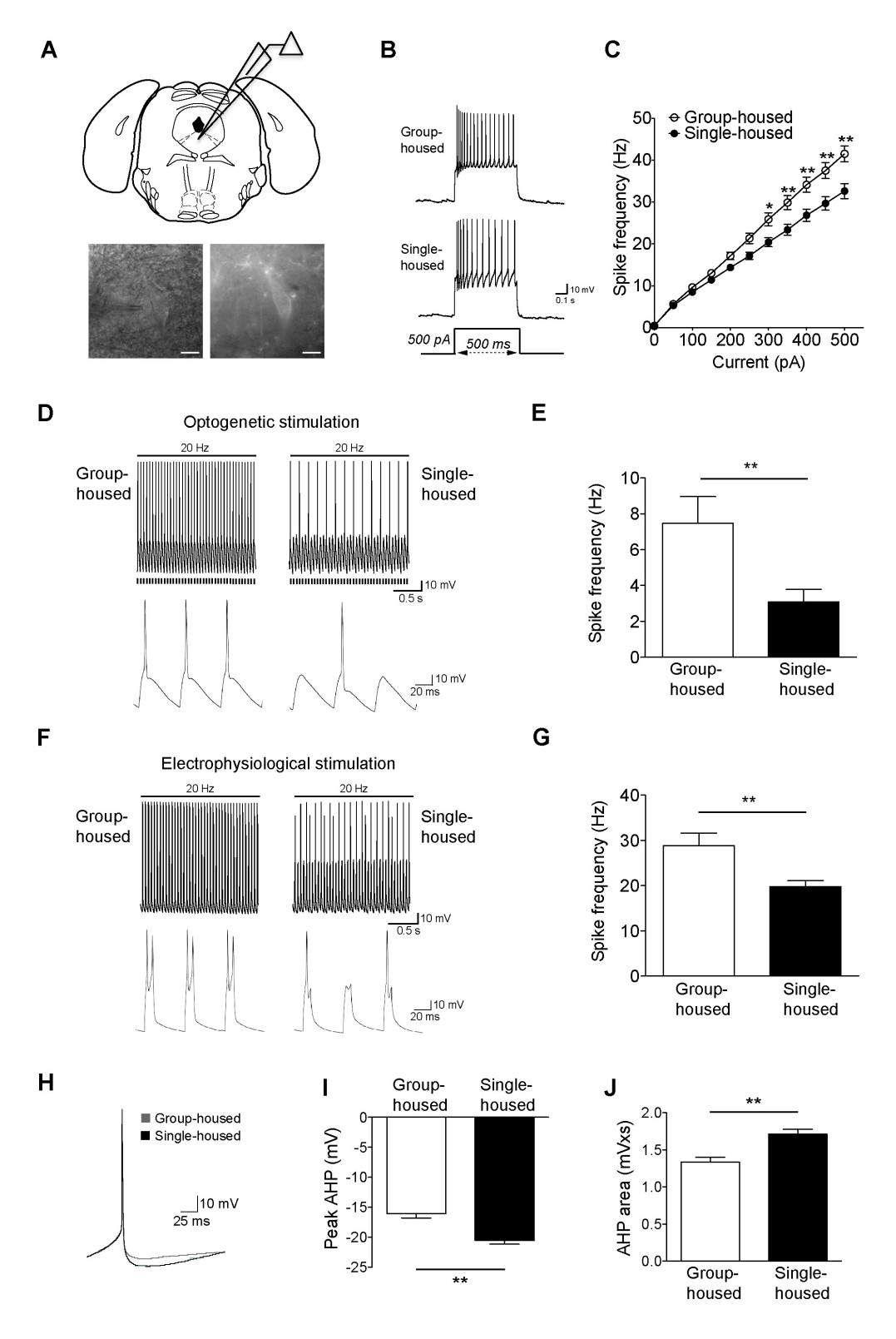

**Figure 1.** Reduced firing of dorsal raphe 5-HT neurons after chronic social isolation is accompanied by an increase in afterhyperpolarization (AHP). (**A**) Schematic representation of a coronal brainstem section (above) comprising dorsal raphe nucleus (adapted from *Paxinos and Franklin, 2001*) where 5-HT neurons are recorded. IR-DIC and EYFP fluorescence images (below) of a dorsal raphe 5-HT neuron being approached by a recording pipette. Scale bars, 10 μM. (**B**) Current-clamp traces of a 5-HT neuron from a group-housed (above) and a single-housed (below) mouse in response to a 500 pA

*Figure 1 continued on next page*

*Figure 1 continued*

depolarizing step. (C) Input-output curve showing spike frequency (Hz) of 5-HT neurons in response to a series of depolarizing current (pA) injections. The firing frequency (Hz) of 5-HT neurons from single-housed mice (n = 38 neurons) is reduced compared to 5-HT neurons from group-housed mice (n = 33 neurons) indicating reduced excitability (two-way repeated-measures ANOVA, effect of housing, $F_{(1690)}$ = 9.19, p=0.003; Newman-Keuls posthoc test, *p<0.05, **p<0.01) (D) Current-clamp recordings of a 5-HT neuron from a group-housed mouse (left, inset below) and a single-housed mouse (right, inset below) showing action potential firing in response to 20 Hz optogenetic stimulation by blue light. Note that the neuron from the group-housed mouse is able to respond with 20 Hz firing frequency to blue light whereas the neuron from the single-housed mouse responds with a 7 Hz firing frequency. (E) The frequency (Hz) of action potentials in response to optogenetic stimulation is reduced in 5-HT neurons from single-housed mice (n = 20 neurons) compared to neurons from group-housed mice (n = 17 neurons) (unpaired *t*-test, *p<0.05). (photocurrents; group-housed: −343 ± 32.3 pA, single-housed: −269 ± 29.4 pA, p=0.1) (F) Current-clamp recordings of a 5-HT neuron from a group-housed mouse (left, inset below) and a single-housed mouse (right, inset below) showing action potential firing in response to 20 Hz electrophysiological stimulation. The neuron from the group-housed mouse produced spike doublets in response to the strong electrophysiological stimulation (left inset). The neuron from the single-housed mouse responded with missing spikes (right inset). (G) The frequency (Hz) of action potentials in response to electrophysiological stimulation is reduced in 5-HT neurons from single-housed mice (n = 14 neurons) compared to neurons from group-housed mice (n = 13 neurons) (unpaired *t*-test, **p<0.01). (H) The first spike of the 25 pA depolarizing step of a current-clamp trace from each 5-HT neuron from group-housed (n = 33 neurons) and single-housed (n = 37 neurons) mice is averaged in order to obtain the resulting trace showing the AHP difference. The peak value of the AHP (mV) (I) and AHP area (mVxs) (J) are greater in 5-HT neurons from single-housed mice, relative to group-housed mice (unpaired *t*-test, **p<0.01). Data are represented as mean ± S.E.M.

The following figure supplements are available for figure 1:

**Figure supplement 1.** Membrane characteristics of dorsal raphe 5-HT neurons from group-housed and single-housed mice are similar.

**Figure supplement 2.** Reduced excitability of 5-HT neurons of single-housed mice is not due to changes in the serotonergic tone.

**Figure supplement 3.** 5-HT neurons in dorsal raphe show reduced firing yet similar inward currents in response to excitatory neuromodulators after chronic social isolation.

**Figure supplement 4.** The reduced firing of 5-HT neurons after chronic social isolation can be normalized by blockers of voltage-gated calcium channels and SK channels, despite the lack of calcium differences.

for regulating their firing activity (*Brown et al., 2001, 2002 Sakurai et al., 2005*; *Tsujino and Sakurai, 2009*). Furthermore, the pacemaker activity of 5-HT neurons in vivo as well as their firing activity on slice is largely dependent on noradrenergic tone (*Baraban et al., 1978*; *Liu et al., 2005*). To examine whether chronic social isolation perturbs the firing activity of DRN 5-HT neurons in response to excitatory neuromodulation, we investigated the effects of orexinB and phenylephrine, which act respectively on hypocretin2 and α1-adrenergic receptors. Application of orexinB or phenylephrine on slice depolarized the majority of 5-HT neurons to threshold and elicited action potentials. Quantification of spike frequency revealed a significant suppression in the peak orexinB (*Figure 1—figure supplement 3A,C*) and the peak phenylephrine effect (*Figure 1—figure supplement 3B,D*) in 5-HT neurons of single-housed mice. These differences in neuromodulator-elicited excitability do not appear to reflect altered efficacy of the receptors nor channel effectors, since upon voltage-clamp examination both neuromodulators produced robust inward excitatory currents comparable between groups (*Figure 1—figure supplement 3E–H*). Taken together, 5-HT neurons from single-housed mice generated significantly reduced action potential frequency in response to stimulation of typical excitatory neuromodulator receptors, suggesting a disturbance in modulation and firing activity in response to excitatory inputs.

## Social isolation increases afterhyperpolarization (AHP) and disturbs the normal molecular balance of SK channels

A key characteristic of 5-HT neurons is their long-duration action potential followed by a medium duration AHP (*Beck et al., 2003*; *Kirby et al., 2003*; *Rouchet et al., 2008*; *Alix et al., 2014*). Since AHP modulation is closely linked to the alterations in the firing activity of 5-HT neurons (*Rouchet et al., 2008*; *Crespi, 2009*), we sought to determine whether the observed reduction in their activity is due to a more pronounced AHP. We found that the peak and area of rheobase AHP

were significantly larger for the single-housed mice (*Figure 1H–J*). The larger AHP in 5-HT neurons after social isolation renders them less excitable and less responsive to stimulation.

Since AHP is dependent on calcium ($Ca^{2+}$) (*Sah, 1996*; *Faber and Sah, 2003*), we examined whether blocking voltage-gated $Ca^{2+}$ channels (VGCC) would modify the firing activity of 5-HT neurons. Using the VGCC blocker $CdCl_2$, the difference in firing frequency of 5-HT neurons from group- and single-housed mice was abolished (*Figure 1—figure supplement 4A–B*). 5-HT neurons from both groups responded with such a robust increase in their firing frequency that they went into a depolarization block at current steps >100 pA (data not shown). As the medium duration AHP in 5-HT neurons has been attributed to SK channels (*Scuvée-Moreau et al., 2004*), we next considered whether we could restore the firing activity of 5-HT neurons from single-housed mice by inhibiting these channels. Using apamin, a blocker of the SK family of channels (*Blatz and Magleby, 1986*; *Köhler et al., 1996*), the firing frequency of 5-HT neurons in response to depolarizing current steps became comparable between groups (*Figure 1—figure supplement 4C–D*). The intracellular $Ca^{2+}$ levels relative to baseline (dF/F) in response to a set number of action potentials were similar between 5-HT neurons of group- and single-housed mice (*Figure 1—figure supplement 4E–F*), indicating that alterations in expression levels and/or sensitivity of SK channels to $Ca^{2+}$ rather than changes in $Ca^{2+}$ responses might contribute to the reduced firing activity of 5-HT neurons upon chronic social isolation.

Two subtypes of SK channels, SK2 ($K_{Ca}2.2$) and SK3 ($K_{Ca}2.3$), exist in the rodent DRN (*Stocker, 2000*). The discrete contribution of each subtype to 5-HT neuron excitability is not known. Both subtypes are sensitive to apamin (*Adelman et al., 2012*); but SK2 can be blocked selectively with a novel toxin Lei-Dab7 (*Shakkottai et al., 2001*; *Aidi-Knani et al., 2015*). With this pharmacological approach, we reveal for the first time a significant contribution of SK2 to the electrophysiological regulation of 5-HT neurons in group-housed mice (*Figure 2A–B*). Yet in single-housed mice, surprisingly, Lei-Dab7 did not produce a significant effect (*Figure 2C–D*). In contrast, blocking both SK2 and SK3 with apamin rendered the excitability of 5-HT neurons comparable in both groups with a significantly greater increase in firing frequency in the single-housed mice (firing frequency to 500 pA step; group-housed: 161.5 ± 6.9% baseline, n = 15 neurons, p<0.001, single-housed: 239.2 ± 16.5% baseline, n = 13 neurons, p<0.001) (*Figure 2E–F*). Western blot analysis in DRN revealed that there was a significant increase in expression of SK3 channels in socially-isolated mice, while the levels of SK2 channels remained unaltered (*Figure 2G*). Finally, apamin application decreased the rheobase AHPs of 5-HT neurons from both groups, making them comparable (*Figure 2H–J*).

## Systemic apamin treatment normalizes anxiety- and depressive-like behaviors in socially isolated mice

To characterize anxiety-/depressive-like behaviors after chronic social isolation, group-housed and single-housed mice were subjected to three different behavioral tests: novelty suppressed feeding, tail suspension, and sucrose preference. To assess the therapeutic potential of inhibiting SK channels for treating depression and anxiety, we systemically administered the SK channel blocker apamin (i. p.) in a group of single-housed mice 30 min before each test.

Single-housed mice showed significantly longer latencies to start feeding in the novelty suppressed feeding test compared to the group-housed mice (*Figure 3A*), indicating elevated anxiety-like behavior after chronic social isolation. Blockade of SK channels by systemic apamin treatment resulted in a trend towards an attenuation of the increased latency to feed (*Figure 3A*). To assess appetitive motivation, we measured the amount of food consumed in the homecage following the novelty suppressed feeding test. Interestingly, single-housed mice consumed significantly more food in the homecage compared to the group-housed mice (*Figure 3—figure supplement 1*). The consumption of food in single-housed mice was restored to control levels following acute systemic apamin treatment (*Figure 3—figure supplement 1*), indicating not only that SK channels may play an important role in feeding behavior, but also that modulation of these channels may be useful in the treatment of stress/anxiety related eating disorders.

In the tail suspension test, chronic social isolation resulted in an increase in depressive-like behavior. Single-housed mice showed significantly increased cumulative immobility (*Figure 3B*). This impairment was completely reversed by acute systemic apamin in single-housed mice (*Figure 3B*). Social isolation also elicited anhedonia, a measure of depressive-like behavior as shown by decreased sucrose preference in single-housed mice, compared to the group-housed mice. SK

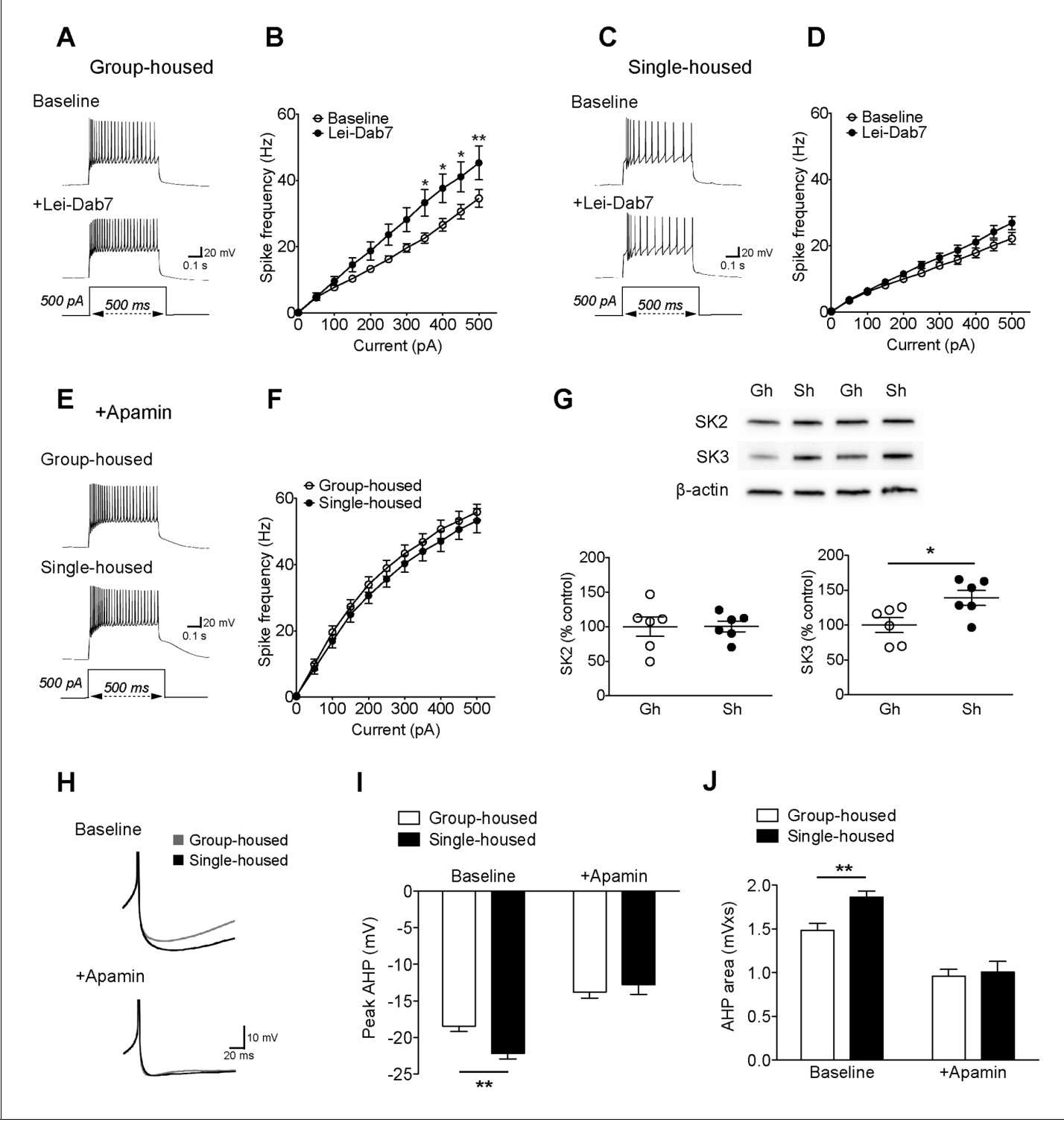

**Figure 2.** The reduced excitability of 5-HT neurons upon chronic social isolation can be restored by blocking SK3 channels. (**A**) Current-clamp traces of a 5-HT neuron in response to a 500 pA depolarizing step before (Baseline, above) and after application of SK2 blocker, Lei-Dab7 (100 nM) (+Lei-Dab7, below) from a group-housed mouse. (**B**) Input-output curve showing increased excitability upon application of Lei-Dab7 in 5-HT neurons of group-housed mice (Baseline, n = 17 neurons, +Lei-Dab7, n = 11 neurons) (two-way repeated-measures ANOVA, effect of drug, $F_{(1260)}$ = 5.87, p=0.023; Newman-Keuls posthoc test, *p<0.05, **p<0.01). (**C**) Current-clamp traces of a 5-HT neuron in response to a 500 pA depolarizing step before (Baseline, above) and after application of SK2 blocker, Lei-Dab7 (100 nM) (+Lei-Dab7, below) from a single-housed mouse. (**D**) The firing frequency (Hz) of 5-HT neurons from single-housed mice did not change upon Lei-Dab7 (Baseline, n = 16 neurons, +Lei-Dab7, n = 9 neurons) (two-way repeated-measures

*Figure 2 continued on next page*

*Figure 2 continued*

ANOVA, effect of drug, $F_{(1230)}$ = 2.39, p=0.13). (**E**) Current-clamp traces of 5-HT neurons in response to 500 pA depolarizing steps after subsequent application of apamin (200 nM) to block SK3 from a group-housed (above) and a single-housed (below) mouse. (**F**) Although the group difference persisted after SK2 blockade with Lei-Dab7, the subsequent application of apamin to block SK3 rendered the firing frequency (Hz) of neurons from group-housed (n = 15 neurons) and single-housed (n = 13 neurons) mice comparable (two-way repeated-measures ANOVA, effect of housing, $F_{(1260)}$ = 0.67, p=0.42). (**G**) Representative immunoblots (top) and quantification (bottom) showing protein levels of SK2 and SK3 channels in the DRN of group- and single-housed mice. SK3 channel expression is significantly increased in single-housed mice (n = 6) compared to group-housed mice (n = 6) (unpaired *t*-test, p=0.027) while SK2 channel expression seems unaltered (unpaired *t*-test, p=0.98). Gh; group-housed, Sh; single-housed. (**H**) Averaged superimposed spikes showing the AHP difference of 5-HT neurons before and after application of apamin in group-housed (Baseline; n = 18, +Apamin; n = 10 neurons) and single-housed (Baseline; n = 16, +Apamin; n = 8 neurons) mice. The peak value of the AHP (mV) (**I**) and AHP area (mVxs) (**J**) before and after application of apamin are shown (two-way ANOVA, effect of drug, peak AHP: $F_{(1,48)}$ = 60.02, p<0.001, AHP area: $F_{(1,48)}$ = 60.59, p<0.001, Newman-Keuls posthoc test, **p<0.01). Data are represented as mean ± S.E.M.

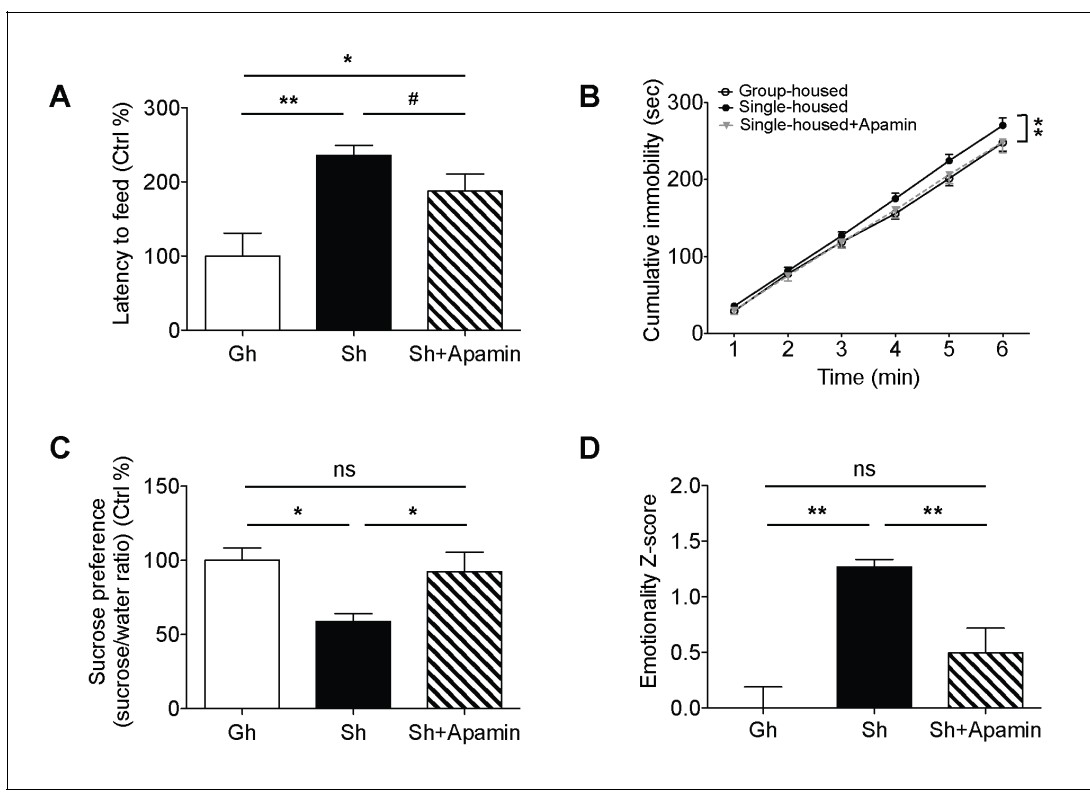

**Figure 3.** Decreased anxiety/depression-like behaviors in single-housed mice after systemic apamin. (**A**) Single-housed mice showed increased latency to feed, which was only partially recovered upon apamin treatment in the novelty suppressed feeding test (one-way ANOVA, group effect, p=0.001, Newman-Keuls posthoc test, *p<0.05, **p<0.01; *a priori* t-test between Sh and Sh+Apamin, #p=0.09). (**B**) Single-housed mice showed increased immobility in the tail suspension test suggesting enhanced depressive-like behavior, normalized by apamin treatment (two-way ANOVA, main effect of group, p=0.01, Newman-Keuls posthoc test, **p<0.01). (**C**) Single-housed mice showed decreased sucrose preference, which was restored by apamin treatment (one-way ANOVA, main effect of group, p=0.02, Newman-Keuls posthoc test, *p<0.05). (**D**) Single-housed mice showed overall increased emotionality, which was decreased by apamin treatment (one-way ANOVA, main effect of group, p<0.01, Newman-Keuls posthoc test, **p<0.01). Data are represented as mean ± S.E.M. (n = 7–9 per group). Gh; group-housed, Sh; single-housed, ns; not significant.

The following figure supplement is available for figure 3:

**Figure supplement 1.** The amount of food consumed in homecage after novelty suppressed feeding test.

channel blockade reduced the anhedonia to levels that were comparable to group-housed mice (*Figure 3C*).

In addition to exploring the behavioral data for each individual test we also combined the z-scores across all behavioral tests to produce an integrated emotionality score for each mouse (*Guilloux et al., 2011*). Integrated z-scores showed that chronic social isolation significantly increased overall behavioral emotionality while acute systemic treatment with apamin was sufficient to decrease emotionality back to control levels (*Figure 3D*).

## Discussion

Our results demonstrate that chronic social isolation results in a reduction in the excitability of 5-HT neurons and renders them less responsive to stimulation. We demonstrate that inhibiting SK channels, critical regulators of AHP in these cells, can reverse the reduced excitability. This work provides the first direct link between a chronic social isolation paradigm and functional alterations in 5-HT neurons themselves. We also reveal that SK2 contributes to normal regulation of 5-HT neuronal excitability but ceases to modulate these neurons significantly after chronic stress. Behavioral analysis showed that chronic social isolation increases anxiety/depression-like behaviors, which can be normalized upon inhibition of SK channels by acute systemic apamin.

In humans, polymorphisms on the gene encoding the SK3 channel have been associated with neuropsychiatric disorders characterized by emotional dysregulation including schizophrenia, bipolar disorder, and anorexia nervosa (*Chandy et al., 1998*; *Koronyo-Hamaoui et al., 2007*; *Grube et al., 2011*). Administration of the SK channel blocker apamin in mice and rats reduced immobility in a forced swim test (*Galeotti et al., 1999*; *van der Staay et al., 1999*), a measure of depressive-like behavior in rodents (*Cryan and Slattery, 2007*). Consistent with these reports, SK3 null mice show enhanced hippocampal 5-HT release and reduced immobility in forced swim and tail suspension tests, indicative of an antidepressant like phenotype (*Jacobsen et al., 2008*). Our work demonstrates for the first time that social isolation, a major risk factor for depression and anxiety, results in reduced 5-HT neuronal activity due to upregulated SK3 channel function in the DRN. Moreover, we show that systemic treatment with apamin, an inhibitor of SK channels improves behavioral deficits induced by chronic social isolation. Consistent with studies that have implicated SK channel function in neuropsychiatric conditions, our current findings suggest that SK channel modulation is a promising therapeutic target for disorders of emotional disturbance such as depression and anxiety. Apamin however has been reported to have adverse side effects in rats (*van der Staay et al., 1999*) close to the beneficial dose in the current mouse study. The diverse expression of SK channels in different tissue types and the lack of tissue/subtype specific modulators or inhibitors are currently the limiting factors for therapeutic interventions targeting SK channels. Further research focused on understanding the differential regulation, modulation and function of SK channel subtypes may shed light on the development of effective disease treatment strategies. Importantly, development of more specific drugs targeted at inhibiting SK3 function may have significant implications for treatment of depressive and anxiety disorders.

## Materials and methods

### Experimental animals

All experiments were performed in Tg(Tph2-COP4*H134R/EYFP)5Gfng mice (RRID:IMSR_JAX: 014555) that express channelrhodopsin EYFP fusion protein driven by the *Tph2* promoter (*Zhao et al., 2011*), in accordance with animal protocols approved by the University of Toronto (20010374, 20011622, 20011733). All mice were maintained in C57Bl/6 background. At weaning (p21), littermate male mice were either housed individually (single-housed) or together in groups of 3–4 (group-housed). After a minimum of 7 weeks, the adult (> p70) mice were used for experiments. Taken together, this study required a total of 28 group-housed and 29 single-housed mice for electrophysiological and western blot experiments, and a total of 25 mice for behavioral experiments. All mice were housed under a 12:12h light/dark cycle with *ad libitum* access to both food and water.

## Electrophysiology

Coronal brainstem slices (400 µm) comprising the dorsal raphe nucleus were obtained using a Dosaka Pro-7 Linear Slicer (SciMedia) in ice-cold oxygenated sucrose-substituted artificial cerebrospinal fluid (ACSF). The slices were then recovered for a minimum of 2 hr in ACSF solution containing: 128 mM NaCl, 10 mM D-glucose, 26 mM $NaHCO_3$, 2 mM $CaCl_2$, 2 mM $MgSO_4$, 3 mM KCl, 1.25 mM $NaH_2PO_4$, pH 7.4 and saturated with 95% $O_2$/5% $CO_2$ at 31–33°C. L-tryptophan (2.5 µM) was included during the recovery period to maintain 5-HT synthesis (*Rood et al., 2014*). Recording was performed in ACSF oxygenated with 95% $O_2$/5% $CO_2$ at 31–33°C flowing at a rate of 3–4 ml/min. Patch pipettes (2–4 MΩ) contained the following composition: 120 mM potassium gluconate, 5 mM KCl, 2 mM $MgCl_2$, 4 mM $K_2$-ATP, 0.4 mM $Na_2$-GTP, 10 mM $Na_2$-phosphocreatine, and 10 mM HEPES buffer (adjusted to pH 7.3 with KOH). Neurons in DRN were visualized with a fixed-staged microscope (Olympus BX50WI) and 5-HT neurons were targeted based on the expression of EYFP. Recordings were focused on the dorsomedial subregion of dorsal raphe, where most of the EYFP positive neurons were visible. Whole-cell recordings were made in voltage-clamp or current-clamp mode with a Multiclamp 700B amplifier (Molecular Devices). Voltage-clamp recordings were performed at −75 mV. Optogenetic stimulation (20 Hz, 10 ms) was performed using LED light (473 nm, Thorlabs). For electrophysiological stimulation, action potentials were elicited by applying brief depolarizing current pulses (>500 pA, 10 ms) at 20 Hz from a membrane potential of −70 mV. All data were acquired at 20 kHz and low-pass filtered at 3 kHz using pClamp10.2 and Digidata1440 software.

## Pharmacology

5-CT (100 nM, Tocris) was bath applied to probe 5-HT$_{1A}$ receptor currents. WAY100635 (30 nM, Sigma) was applied in ACSF to block 5-HT$_{1A}$ receptor responses. OrexinB (300 nM, GenScript) and R-(-) Phenylephrine hydrochloride (10 µM, Sigma) were applied in bath to measure responses of 5-HT neurons to excitatory neuromodulation. Picrotoxin (100 µM, Sigma) and CGP52432 (1 µM, Tocris) were included together in bath solution to block GABA$_A$ and GABA$_B$ receptor responses, respectively. Cadmium chloride (100 µM, Sigma) was used to block voltage gated calcium channels. Lei-Dab7 (100 nM, Tocris) was included in ACSF as a blocker of SK2 channels. Apamin (200 nM, Tocris) was used as a blocker of SK2 and SK3 channels.

## Multiphoton calcium imaging

The calcium dye Fluo5F (150 µM, Invitrogen) was included in the pipette, along with Alexa Fluor-594 hydrazide (20 µM), for visualization of the neuron. Multiphoton imaging was performed using a titanium:sapphire laser (Newport) tuned to wavelength 800 nm and an Olympus Fluoview FV1000 microscope equipped with a 60X objective (numerical aperture 0.9). Images were acquired at a rate of ~10 frames/s. Emissions were recorded in the green channel (Fluo5F signal, 495–540 nm) and the red channel (Alexa Fluor-594 signal, 570–620 nm). Analysis was performed using Fluoview software by selecting a pan-somatic area of interest and measuring the increases in green fluorescence relative to baseline fluorescence (dF/F).

## Western blot

For Western blot analysis, brainstem slices were prepared in the same way as for electrophysiological experiments. DRN was then dissected and immediately frozen on dry ice. The tissue was lysed in lysis buffer (Biobasic) supplemented with protease and phosphatase inhibitors. Protein concentration was measured using Bradford assay (Sigma). For immunoblotting, 10 µg protein was loaded per lane onto 4–20% Mini-Protean precast polyacrylamide gels (Bio-Rad), separated by electrophoresis and transferred onto a nitrocellulose membrane (Bio-Rad). Membranes were then blocked with 5% non-fat milk in Tris-buffered saline and incubated overnight at 4°C with primary antibodies (rabbit anti-SK3, 1:1000, Alomone, RRID:AB_2040130; rabbit anti-SK2, 1:1000, Alomone, RRID:AB_2040126; rabbit anti-5-HT$_{1A}$R, 1:1000, Millipore Bioscience Research Reagents, RRID:AB_805421; mouse anti-β-actin, 1:5000, Sigma, RRID:AB_476744). Immunoreactive bands were visualized using HRP-coupled secondary antibodies by ECL detection methods (Bio-Rad). Band intensities were analyzed using Bio-Rad Image Laboratory software and calculated relative to β-actin as the loading control.

## Behavioral analysis

To assess anxiety- and depression-like behaviors, the novelty suppressed feeding, tail suspension, and sucrose preference tests were performed. Each test was separated by a minimum of 3 days. Mice were habituated to the testing room for at least 30 min. Apamin (0.3 mg/kg dissolved in 0.9% NaCl, Alomone Labs) or vehicle (0.9% NaCl) was injected intraperitoneally (i.p.) 30 min before each test (*van der Staay et al., 1999*; *Chen et al., 2014*).

### Novelty suppressed feeding test

Following food deprivation for 18 hr, mice were provided with a single food pellet placed in the middle of a dimly lit plastic (45 x 45x 20 cm) box. The floor of the box was covered with bedding 2 cm deep. The latency to start feeding in a 5 min assay was measured by an observer blinded to the groups/treatment. Following the test, each mouse was returned to its homecage and the amount of food consumed within 5 min was measured.

### Tail suspension test

Mice were individually suspended by their tails within a three-walled plastic box (45 x 25 x 25 cm) using adhesive tape securely attached to the tip of the tail. The test lasted for 6 min and the behavior of each mouse was videotaped. The amount of time spent immobile per minute was measured by an observer blinded to the groups and treatment and expressed as the cumulative immobility.

### Sucrose preference test

Mice were habituated to drinking 2% sucrose from two bottles prior to the test. In order to assess the drinking amount, group-housed mice were housed in pairs or singles for a period of 3 days. Mice were given a choice of 2% sucrose or water in their homecage for 2 days. Water and sucrose intake were measured daily and the position of the bottles were changed after the first day. Drugs were administered on both days of the test. Data were expressed as the ratio of sucrose to water consumed and the average of both days were taken.

Z-scoring methodology was used as previously described (*Guilloux et al., 2011*) to assess the overall emotionality score for each mouse that ran through all the behavioral tests.

## Data analysis

Electrophysiological responses were calculated as changes in membrane potential or holding current resulting from manipulations such as 473 nm light exposure or the bath application of drugs and analyzed by Clampfit (Molecular Devices) software. Spike frequencies for input-output curves were measured by counting the number of spikes in response to 500 ms depolarizing current steps. Spike frequencies for optogenetic and electrophysiological stimulations were determined by counting the number of spikes over a 2 s stimulation period. For illustrative purposes, averaged traces showing current responses and AHPs were compiled using Axograph software. Quantification of AHP potentials was performed by detecting action potentials automatically with a derivative threshold of 20 mV/ms by Axograph software. AHP area (mVxs) is measured as the area under the AHP between the midpoint of each spike (taken as time=0) and the 100 ms time point corresponding to the medium AHP (*Sah, 1996*; *Faber and Sah, 2003*). The maximum negative value of the AHP within this time window is determined as the peak AHP (mV). Statistical analysis was performed using Prism 5 software. For electrophysiology experiments, the numbers (n) that represent the number of neurons were obtained from at least 3 mice per group. For western blots, numbers (n) represent the number of mice (averages of 2 replicates per mouse). Data were analyzed using unpaired student's *t*-test, two-way ANOVA or repeated-measures two-way ANOVA. Newman-Keuls post-hoc test was performed when appropriate. Data were expressed as mean ± S.E.M. and evaluated at a significance level of 0.05.

## Acknowledgements

This work was funded by grants to EKL from the Canadian Natural Science and Engineering Research Council (NSERC), the Canada Research Chairs Program, and the Early Researcher Award from the Province of Ontario. We thank Ms. Lily Kang and Ms. Rhian Duke for expert technical assistance.

# Additional information

## Funding

| Funder | Grant reference number | Author |
|---|---|---|
| Natural Sciences and Engineering Research Council of Canada | Discovery Grant | Evelyn K Lambe |
| Canada Research Chairs | Canada Research Chair in Developmental Cortical Physiology | Evelyn K Lambe |
| Province of Ontario, Canada | Early Researcher Award | Evelyn K Lambe |

The funders had no role in study design, data collection and interpretation, or the decision to submit the work for publication.

## Author contributions

DS, Conception and design, Acquisition of data, Analysis and interpretation of data, Drafting or revising the article; DKO, Acquisition of data, Analysis and interpretation of data, Drafting or revising the article; EKL, Conception and design, Analysis and interpretation of data, Drafting or revising the article

## Author ORCIDs

Derya Sargin, http://orcid.org/0000-0002-0253-5442
David K Oliver, http://orcid.org/0000-0003-1210-8409
Evelyn K Lambe, http://orcid.org/0000-0002-5994-6090

## Ethics

Animal experimentation: This study was performed in accordance with the recommendations of the Canadian Council on Animal Care. All of the animals were handled according to approved institutional animal care and use guidelines under protocols approved by the Faculty of Medicine Animal Care and Use Committee at the University of Toronto. (20010374, 20011622, 20011733)

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
