## [Decision Letter]

[Editors’ note: a previous version of this study was rejected after peer review, but the authors submitted for reconsideration. The first decision letter after peer review is shown below.]

Thank you for submitting your work entitled "Chronic social isolation reduces 5-HT neuronal activity via upregulated SK3 calcium-activated potassium channels" for consideration by *eLife*. Your article has been reviewed by two peer reviewers, one of whom, Sacha B Nelson, is a member of our Board of Reviewing Editor and the evaluation has been overseen by Richard Aldrich as the Senior Editor.

Our decision has been reached after consultation between the reviewers. Based on these discussions and the individual reviews below, we regret to inform you that your work will not be considered further for publication in *eLife*.

Although the topic of the experiments is interesting and the results presented are convincing, the reader is left unsure of the behavioral impact of the changes in excitability observed and as to whether this is indeed as suggested a "promising therapeutic target for disorders of emotional disturbance such as depression and anxiety." It remains possible that SK channels are too broad a target to be useful, and even that these changes in excitability are consequences of, but not causally related to, the effects of social deprivation. For example, one reviewer noted (during the reviewer discussion) that "a more recent paper used a somewhat similar model in mice (Neuropsychopharmacology. 2014 Dec; 39(13):2928-37. doi: 10.1038/npp.2014.162) and doesn't seem to see much on the anxiety tests; they did voltammetry measures of 5-HT release in different conditions, and rather see a trend for increased rather than decreased 5-HT release (see their table1). " While it would be feasible with significant additional work to shore up the present results with additional behavioral studies (e.g. demonstrating that social isolation produces specific behavioral effects which can be normalized by normalizing the excitability of DRN neurons), this would likely take several months to complete and it is the journal policy to reject manuscripts which cannot be accepted without work that could be completed in 1-2 months at most. In this case, however, we would be willing to consider a new manuscript on the same topic, provided it addressed the issue of behavioral relevance. But the reviewer and reviewing editor felt that in its present state the manuscript would be more appropriate for a more specialized journal.

Reviewer #1:

The authors show that social isolation results in reduced excitability of serotonergic neurons in the mouse dorsal raphe, perhaps explaining the impact of social isolation on depression related symptoms. The experiments demonstrate concisely and convincingly that the decreased excitability is due to an increased afterhyperpolarization (AHP) associated with increased expression of the SK3 subtype of small conductance Ca-activated potassium (SK) channels. Interestingly, polymorphisms of the SK3 gene are known to be risk factors for psychiatric disease.

The paper is short and to the point, and the experiments are well done and clearly illustrated.

I had only minor suggestions for clarifying the physiological results presented in the manuscript. However, after discussion with the other reviewer, I am persuaded that there is still an open question as to the behavioral relevance, and hence the general interest, of the results. Although it is clear that social isolation is a risk factor for depressive symptoms and anxiety, it is not clear whether the observed change in DRN excitability plays a causal role in this.

Reviewer #2:

The work nicely demonstrates that rearing mice in social isolation lowers the excitability of serotonin raphe neurons, by modifying the afterhyperpolarization (AHP) responses of the 5-HT neurons, a biophysical property that is controlled by SK channels; they find that SK3 channel expression is increased in the raphe of single-reared mice and that pharmacological blockade of SK3 normalizes AHP and raphe neuron excitability. The paper is well done, but without addition of behavioral data, it remains limited in its conclusions and does not address the possible behavioral correlates of SK3 changes (nor the possible upstream mechanisms that induce SK3 up regulation). Thus, although presenting an interesting novel observation in the field, the results seemed somewhat too preliminary to conclude on the potential relevance of SK3 up regulation for the effects of stress on mood disorders.

Specific remarks

1) Antagonist rescues hypoactivity of the 5-HT neurons. What about behaviour? There are no behavioural data to show that the protocol of social isolation really changes behaviour, nor how SK3 antagonism could modify them. This is difficult to predict from the published literature since SK3 overexpression was found to cause learning deficits but no anxiety phenotypes (Grube et al., 2011), while SK3-KO appear to have increased anxiety (Jacobsen et al., 2008).

2) Increased of SK3 is shown with western blots, however this shows an unequal signal among cases; thus dot plots would be preferable to histograms in Figure 2. Further, SK3-immunolabelling (such antibodies are characterized in Jacobsen et al.2008) would be a nice addition, to show more specifically an SK3 increase in the 5-HT neurons.

3) In the Results section the authors claim that difference in excitability did not result from changes in GABA tone mentioning data not shown. It would be important to show these data since there is literature showing that stress affects excitability of GABA neurons but not 5-HT neurons (e.g. Challis et al., 2013).

4) When measuring AHP amplitudes and areas, at which frequency were the cells firing (Figure 1)? The authors report the same current step for all cells (25pA); because neurons from single-housed are hypoexcitable we can infer that the frequency is lower. This would bias the measures. Measures of medium AHP are normally taken after a depolarizing current step has been applied, and selecting the current step showing the same number of action potentials for all cells.

5) Because of the heterogeneity of the dorsal raphe neurons, the authors need to specify more precisely the localisation of their recordings.

[Editors’ note: what now follows is the decision letter after the authors submitted for further consideration.]

Thank you for submitting your article "Chronic social isolation reduces 5-HT neuronal activity via upregulated SK3 calcium-activated potassium channels" for consideration by *eLife*. Your article has been reviewed by two peer reviewers: Sacha B Nelson (Reviewer #1), who is a member of our Board of Reviewing Editors, and Patricia Gaspar (Reviewer #2). The evaluation has been overseen by Richard Aldrich as the Senior Editor.

The reviewers have discussed the reviews with one another and the Reviewing Editor has drafted this decision to help you prepare a revised submission.

Summary:

The authors nicely demonstrate that rearing mice in social isolation lowers the excitability of serotonin raphe neurons, by modifying their afterhyperpolarizations (AHPs), a biophysical property that is controlled by small-conductance Ca^2+^-activated potassium (SK) channels. They go on to show that isolation also impacts behaviors associated with anxiety and depression in rodents and that these changes are reversed by inhibiting SK channels.

Essential revisions:

The requested revisions are minor and this should not need to go back for another round of review. The full text of the reviews is given below.

Reviewer #1:

In this revised manuscript, the authors have now added three behavioral tests to support the idea that the changes in DRN SK3 activity they see following social isolation has effects on feeding and the anxiety/depression related behavior of immobility following tail suspension. The results and some other minor fixes add confidence that the observed changes in the excitability of serotonergic neurons are more likely to have a behavioral impact. There is some concern of course that changes elsewhere (e.g. in the gut) could have affected the feeding behaviors, but this concern is difficult to address.

Reviewer #2:

The authors have addressed my previous concerns concerning the validation of the social isolation model in mice, by adding behavioral experiments that demonstrate the effectiveness of their protocol to induce anxiety-like phenotypes. Moreover they indicate a causal link between SK3 upregulation and these phenotypes by showing a partial rescue of the behavioral phenotype with Apamin. Finally methodological precisions and complementary explanations were added to the text as requested.

Minor comments:

1) In their rebuttal, the authors explain the difference of their results with the Dankovski 2014 paper as a difference in protocol. However all the other supporting papers they cite have been done in rats (and they do not cite the Dankovski paper). If this is the first time the social separation model could be valuably transferred to mice, this would be worth mentioning more explicitly for future studies in mice.

2) In this regard, information about the mouse strain/gender of mice used in the different experiments is important; I could not find this info in the new version of the manuscript.

3) Results, subsection “Social isolation reduces excitability of 5-HT neurons”, first paragraph: explain what picrotoxin and CGP52432 are doing (as in the Methods).

4) Results, subsection “Systemic apamin treatment normalizes anxiety- and depressive-like 160 behaviors in socially isolated mice”, last paragraph: to see how the z-score was calculated, I went to "Lin and Sibille 2015"; this was a waste of time as the Methods only refers to Guilloux 2011. Suggestion is to keep only the useful reference.

5) Conclusion - It is interesting that SK channels are a druggable compound for depression. Since this is not entirely new, as it had already been put forth and tested - to some extent - by Crespi in 2010, it may be useful to add a sentence as to why no one has really followed this up. Are there issues of toxicity? A need for more selective agents?

---

## [Author Response]

[Editors’ note: the author responses to the first round of peer review follow.]

*[…] Although the topic of the experiments is interesting and the results presented are convincing, the reader is left unsure of the behavioral impact of the changes in excitability observed and as to whether this is indeed as suggested a "promising therapeutic target for disorders of emotional disturbance such as depression and anxiety." It remains possible that SK channels are too broad a target to be useful, and even that these changes in excitability are consequences of, but not causally related to, the effects of social deprivation. For example, one reviewer noted (during the reviewer discussion) that "a more recent paper used a somewhat similar model in mice (Neuropsychopharmacology. 2014 Dec; 39(13):2928-37. doi: 10.1038/npp.2014.162) and doesn't seem to see much on the anxiety tests; they did voltammetry measures of 5-HT release in different conditions, and rather see a trend for increased rather than decreased 5-HT release (see their table1). " While it would be feasible with significant additional work to shore up the present results with additional behavioral studies (e.g. demonstrating that social isolation produces specific behavioral effects which can be normalized by normalizing the excitability of DRN neurons), this would likely take several months to complete and it is the journal policy to reject manuscripts which cannot be accepted without work that could be completed in 1-2 months at most. In this case, however, we would be willing to consider a new manuscript on the same topic, provided it addressed the issue of behavioral relevance. But the reviewer and reviewing editor felt that in its present state the manuscript would be more appropriate for a more specialized journal.*

We appreciate that the editors and reviewers brought a recent paper into discussion that uses a different model of social isolation and employed voltammetry to address questions with respect to serotonin release. This gives us an opportunity to talk about our model of social isolation in a more detailed manner. We isolate mice at the age of postnatal (P) day 21 upon weaning and this social isolation continues for more than 7 weeks. Our behavioral work in the new Figure 3 demonstrates that this paradigm elicits robust anxiety- and depressive-like effects.

Indeed, it has been shown in various studies that there is a critical developmental window for isolation rearing experiments in rodents and that single-housing after this critical developmental window does not lead to the same robust behavioral effects (Einon and Morgan, 1977; Fone and Porkess, 2008).

In the above-mentioned paper by Dankoski et al. 2016, mice were socially isolated later, starting at adolescence (P35-42), for a relatively brief period of 3 weeks. In this model, single-housed mice did not show robust behavioral changes compared to the paired controls. Although single- housing mice in adolescence may certainly have stressful effects (Wallace et al., 2009), the Dankoski et al. model likely represents a milder stress than ours. Still interestingly, citalopram-induced facilitation of serotonin release was blocked in this model of social isolation suggesting that even milder stressful conditions may have an impact on serotonin release under challenge.

Our electrophysiological findings that serotonin neurons become less responsive to stimulation upon chronic social isolation beginning at P21 are consistent with previous reports showing decreased release of 5-HT in postsynaptic areas after single-housing rodents starting at this developmentally-sensitive time (Jaffe et al., 1993; Bickerdike et al., 1993; Heidbreder et al., 2000; Muchimapura et al., 2002; Muchimapura et al., 2003).

*Reviewer #1:*

*[…] I had only minor suggestions for clarifying the physiological results presented in the manuscript. However, after discussion with the other reviewer, I am persuaded that there is still an open question as to the behavioral relevance, and hence the general interest, of the results. Although it is clear that social isolation is a risk factor for depressive symptoms and anxiety, it is not clear whether the observed change in DRN excitability plays a causal role in this.*

We thank the reviewer for raising the point with respect to the behavioral relevance of our findings. We have now performed the behavioral experiments showing that this chronic social isolation paradigm elicits significant differences on anxiety- and depressive-like behaviors.

Furthermore, we demonstrate that this group difference can be rescued with SK channel blockade by apamin used at a dose shown to enhance the excitability of raphe neurons in vivo (Crespi, 2009).

*Reviewer #2:*

*[…] Specific remarks*

*1) Antagonist rescues hypoactivity of the 5-HT neurons. What about behaviour? There are no behavioural data to show that the protocol of social isolation really changes behaviour, nor how SK3 antagonism could modify them. This is difficult to predict from the published literature since SK3 overexpression was found to cause learning deficits but no anxiety phenotypes (Grube et al., 2011), while SK3-KO appear to have increased anxiety (Jacobsen et al., 2008).*

We appreciate the valuable feedback with respect to the requirement of behavioral experiments. We used male mice for our experiments. Jacobsen et al., 2011 reported an antidepressant-like phenotype in SK3 deficient male mice (reduced immobility in tail suspension and forced swim tests), which supports our findings in this study.

With the suggestion of the reviewer, we now conducted the necessary behavioral experiments and added them into the manuscript. A new Figure 3 and Figure 3—figure supplement 1 highlight the striking behavioral differences observed in our chronically socially isolated male mice compared to group- housed littermate controls, as well as the normalization of these measures upon acute administration of apamin to block a portion of SK channels.

Interestingly, Jacobsen et al., 2011 reported an anxiety-like effect in one of the tests (zero maze) in SK3 deficient female mice. Whether sex differences exist in our model is an interesting point and will be important to follow up in a future study.

*2) Increased of SK3 is shown with western blots, however this shows an unequal signal among cases; thus dot plots would be preferable to histograms in Figure 2. Further, SK3-immunolabelling (such antibodies are characterized in Jacobsen et al. 2008) would be a nice addition, to show more specifically an SK3 increase in the 5-HT neurons.*

We now included the dot plots instead of the histograms for the western blot data in Figure 2. SK3 expression is reported to be predominantly in serotonergic neurons of dorsal raphe (Stocker and Pedarzani, 2000). Serotonergic neurons characteristically have a prominent medium duration apamin-sensitive AHPs (Scuvée-Moreau et al., 2004) that do not exist to the same extent in other neuronal subtypes such as GABAergic neurons in dorsal raphe. Moreover, systemic apamin injections were shown to increase the firing rate of 5-HT neurons in vivo and resulted in a subsequent increase in 5-HT release in postsynaptic regions (Crespi, 2009) suggesting that serotonergic neurons are specifically sensitive to apamin. SK channel blockade in vivo has also been shown to increase burst firing specifically in serotonergic neurons (Rouchet et al., 2008; Crespi, 2009). These effects of apamin persisted in the presence of GABAA and GABAB receptor blockers (Rouchet et al., 2008). Together with our work, these experiments suggest that serotonergic neurons would contribute disproportionately and be most sensitive to alterations in the expression of SK channels in this region.

*3) In the Results section the authors claim that difference in excitability did not result from changes in GABA tone mentioning data not shown. It would be important to show these data since there is literature showing that stress affects excitability of GABA neurons but not 5-HT neurons (e.g. Challis et al., 2013).*

We thank the reviewer for raising this important point. We have now included these data in Figure 1—figure supplement 1. Blockade of GABA-A and GABA-B receptors respectively with picrotoxin and CGP52432 did not attenuate the significant difference in the responsiveness of serotonin neurons between the single- and group-housed conditions. Note, it is essential to use picrotoxin instead of bicuculline for these experiments, since the latter has a known nonspecific effect of blocking SK channels (Khawaled et al., 1999).

*4) When measuring AHP amplitudes and areas, at which frequency were the cells firing (Figure 1)? The authors report the same current step for all cells (25pA); because neurons from single-housed are hypoexcitable we can infer that the frequency is lower. This would bias the measures. Measures of medium AHP are normally taken after a depolarizing current step has been applied, and selecting the current step showing the same number of action potentials for all cells.*

To measure AHP amplitudes and areas, we selected the minimal depolarizing current step required to elicit action potentials in these cells. We calculated the AHPs from the first spike generated in response to the rheobase current. The frequency of firing at this point is similar between the groups (group-housed; 5.8 ± 0.2 Hz, single-housed; 5.4 ± 0.2 Hz, two-tailed t-test p = 0.2). Of note, we found the frequency differences between the groups in response to stronger depolarizing current steps (>250pA).

*5) Because of the heterogeneity of the dorsal raphe neurons, the authors need to specify more precisely the localisation of their recordings.*

We now included this in the Materials and methods section.

[Editors' note: the author responses to the re-review follow.]

*[…] Essential revisions:*

*The requested revisions are minor and this should not need to go back for another round of review. The full text of the reviews is given below.*

*[…] Reviewer #2:*

*1) In their rebuttal, the authors explain the difference of their results with the Dankovski 2014 paper as a difference in protocol. However all the other supporting papers they cite have been done in rats (and they do not cite the Dankovski paper). If this is the first time the social separation model could be valuably transferred to mice, this would be worth mentioning more explicitly for future studies in mice.*

We thank the reviewer for noticing the requirement of additional references for mouse studies on social isolation. We now included additional references in the Introduction.

*2) In this regard, information about the mouse strain/gender of mice used in the different experiments is important; I could not find this info in the new version of the manuscript.*

We used only male mice in our study. The gender and the background of our mice are included in the Methods section.

*3) Results, subsection “Social isolation reduces excitability of 5-HT neurons”, first paragraph: explain what picrotoxin and CGP52432 are doing (as in the Methods).*

*We added this information to this subsection.*

*4) Results, subsection “Systemic apamin treatment normalizes anxiety- and depressive-like 160 behaviors in socially isolated mice”, last paragraph: to see how the z-score was calculated, I went to "Lin and Sibille 2015"; this was a waste of time as the Methods only refers to Guilloux 2011. Suggestion is to keep only the useful reference.*

*The Lin and Sibille 2015 reference is no longer included.*

*5) Conclusion - It is interesting that SK channels are a druggable compound for depression. Since this is not entirely new, as it had already been put forth and tested - to some extent - by Crespi in 2010, it may be useful to add a sentence as to why no one has really followed this up. Are there issues of toxicity? A need for more selective agents?*

We thank the reviewer for raising this important point. We now included in the Discussion what additional steps/approaches are required for targeting SK channels for treatment of depression. Including these points also gave us a chance to explain the translational aspect of our study with the goal of finding better treatment strategies for depression and anxiety.